# Pathogenic and genomic characterization of rabbit-sourced *Pasteurella multocida* serogroup F isolates recovered from dead rabbits with respiratory disease

Jinxiang Wang,[1,2] Shikun Sun,[1,2] Dongjin Chen,[1,2] Chenfang Gao,[1,2] Lei Sang,[1,2] Xiping Xie[1,2]

**ABSTRACT**  *Pasteurella multocida* serogroup F can infect a number of animals. However, the pathogenicity and genomic features of this serogroup are still largely unknown. In the present study, the pathogenicity and genomic sequences of 19 rabbit-sourced *P. multocida* serogroup F isolates were determined. The 19 isolates were highly pathogenic for rabbits causing severe pathologic lesions and high mortality in inoculated rabbits. Nevertheless, the pathologic lesions in rabbits caused by the 19 isolates were distinct from those caused by the previously reported high-virulent serogroup F strains J-4103 (rabbit), P-4218 (turkey), and C21724H3km7 (chicken). Moreover, the 19 isolates were avirulent to white feather broilers. The genomes of the 19 isolates were determined to understand the pathogenicity of these isolates. The finding of a number of functional genes in the 19 isolates by comparison with the low-virulent rabbit-sourced serogroup F strain s4 might contribute to the high virulence of these isolates. Notably, polymorphisms were determined in the lipopolysaccharide outer core biosynthetic genes *natC* and *gatF* among the serogroup F strains of different hosts. However, the sequences of *natC* and *gatF* from rabbit-sourced strains (except for SD11) were identical, which might be responsible for the host specific of the 19 isolates. The observations and findings in this study would be helpful for the understanding of the pathogenicity variation and host predilection of *P. multocida*.

**IMPORTANCE**  The 19 rabbit-sourced *Pasteurella multocida* serogroup F isolates showing high virulence to rabbits were avirulent to the broilers. Notably, polymorphisms were determined in the lipopolysaccharide outer core biosynthetic genes *natC* and *gatF* among all serogroup F strains of different hosts. However, the sequences of *natC* and *gatF* from rabbit-sourced strains (except for SD11) were identical, which might be responsible for the host specific of the 19 isolates.

**KEYWORDS**  rabbit, *Pasteurella multocida* serogroup F, pathogenicity, whole-genome sequence

*P*asteurella multocida is an important bacterial pathogen that can infect a number of animals (1). *P. multocida* is responsible for a variety of economically important diseases such as fowl cholera of poultry, hemorrhagic septicemia of bovine, progressive atrophic rhinitis and respiratory disease complex of swine, and rhinitis and pneumonia of rabbits (1, 2).

  *P. multocida* strains can be grouped into five capsular serogroups (A, B, D, E, and F) based on the capsular polysaccharide antigens (3). *P. multocida* serogroup F was first isolated from turkeys in the USA in 1987 (4). For a long time, this serogroup has been mainly found in avian hosts and recognized as the causative agent of fowl cholera (5–7). Nevertheless, detections of this serogroup in swine (8), sheep (9), cattle (10), and

Address correspondence to Jinxiang Wang, wjx841227@126.com, or Xiping Xie, xxp702@163.com.

The authors declare no conflict of interest.

See the funding table on p. 13.

rabbits (11, 12) with different manifestations have also been reported over the years. The detection of *P. multocida* serogroup F in rabbits was first documented in 2004 (13), and the highly pathogenic of this serogroup for rabbits was confirmed in 2008 (14).

Fujian, in the southeast of China, is an important rabbit farming area of China. Our previous work showed that *P. multocida* was one of the most important causative agents causing significant morbidity and mortality of rabbits in Fujian, whereas only *P. multocida* strains of serogroups A and D were recovered from the lungs of dead rabbits with respiratory disease (15). To our knowledge, the detection of rabbit-sourced *P. multocida* serogroup F in Fujian was uncommon before 2020 (16). Since then, the presence of rabbit-sourced *P. multocida* serogroup F has been detected in different regions of Fujian. Recent reports also showed that rabbit-sourced *P. multocida* serogroup F was widespread in rabbits geographically such as Spain (11), Portugal (11), Italy (12), and China (17). However, the information on the pathogenicity and genetic characteristics of *P. multocida* serogroup F is still limited. In this study, the capsular and lipopolysaccharide (LPS) genotypes, sequence types, and virulence gene profiles of 19 rabbit-sourced *P. multocida* serogroup F isolates recovered from the lungs of dead rabbits with respiratory disease were defined, and the pathogenicity of these isolates was evaluated. Furthermore, the genomes of the 19 isolates were also determined, and the comparative genome analyses between these isolates and other *P. multocida* strains were performed. This study aimed to determine the pathogenicity and genetic characteristics of the 19 rabbit-sourced *P. multocida* serogroup F isolates.

## MATERIALS AND METHODS

### Bacterial isolates

In all, 19 rabbit-sourced *P. multocida* serogroup F isolates (PF1–PF19) were used in this study. All of the 19 isolates were recovered from the lungs of dead rabbits with respiratory disease, and each of the isolates was recovered from a separate rabbit farm (Table S1). The isolate was plated on the brain heart infusion (BHI) agar plate containing 5% defibrinated sheep blood and incubated at 37°C for 24 h. A single bacterial colony of the isolate was picked up and inoculated into 5 mL of BHI containing 2% bovine serum, which was shaken at the conditions of 180 rpm and 37°C for 16 h. For the bacterial inoculation, the bacterial cells were harvested by centrifugation, the cell pellet was washed with sterile normal saline three times, and the bacterial cells were suspended in sterile normal saline reaching a final concentration of $6.0 \times 10^5$ colony forming units (CFU) per milliliter.

### Capsular and LPS genotyping

The genomic DNA of the isolate was prepared using the EasyPure Bacteria Genomic DNA Kit (TransGen Biotech, Beijing, China), and the capsular and LPS genotyping of the isolate was performed using the multiplex PCR assays developed by Townsend et al. (3) and Harper et al. (18) respectively. The products of the multiplex PCR assays were purified and sequenced, and the identities of the products were confirmed by comparing the sequences against the NCBI database.

### Multi-locus sequence typing

The sequence types of the isolates were determined using the multi-host multi-locus sequence typing (MLST) scheme. The housekeeping genes of the isolate were amplified by PCR assays as described in the PubMLST database (https://pubmlst.org/pmultocida/). The PCR products were then purified and sequenced. The allelic numbers of the housekeeping genes were defined by comparing the sequences of the housekeeping genes against the PubMLST database, and the sequence type of the isolate was defined according to the profile of the allelic numbers. The housekeeping genes of the

multi-host MLST scheme of the isolate were concatenated, and then the neighbor-joining phylogenetic tree (1,000 bootstrap replications) was constructed using MEGA 5.0.

## Virulence genes determination

A total of 26 virulence genes were screened in the genomes of the 19 isolates using BLAST (version 2.4.0) with sequence identity of ≥85% and an alignment coverage of ≥85%. These virulence genes including adhesion-related proteins (*fimA*, *hsf-1*, *hsf-2*, *pfhA*, *pfhB1*, *pfhB2*, *ptfA,* and *tadD*), dermonecrotoxin (*toxA*), iron-binding proteins (*exbB*, *exbD*, *fur*, *hgbA*, *hgbB*, *tbpA,* and *tonB*), sialidases (*nanB* and *nanH*), hyaluronidase (*pmHAS*), outer membrane proteins (*ompA*, *ompH1*, *ompH2*, *ompW,* and *oma87*), and superoxide dismutase (*sodA* and *sodC*).

## Antimicrobial susceptibility test

The antimicrobial susceptibility of the 19 rabbit-sourced *P. multocida* was determined using the disk diffusion method according to the Clinical and Laboratory Standards Institute guidelines (19). Ten antibiotics with the indicated content (µg/disk) were used: ampicillin (10), ceftriaxone (20), cefotaxime (20), clarithromycin (15), telithromycin (15), tetracycline (20), ciprofloxacin (5), ofloxacin (5), chloramphenicol (20), and rifampin (5). The results were interpreted according to the interpretive standards for *Haemophilus influenza* and *Haemophilus parainfluenzae*. The *H. influenzae* strain ATCC 49247 was used as the quality control.

## Animal experiments

Rabbits and white feather broilers were used to determine the pathogenicity of the isolates. The rabbits and white feather broilers were purchased from local farms. Before infection, nasal, conjunctival, and rectal swabs of the rabbits as well as oropharyngeal and cloacal swabs of the broilers were collected for bacteriological examination to check the *P. multocida* status of the animals (14). The sera of the rabbits and broilers were also collected to check the presence of the *P. multocida* IgG using an indirect ELISA assay as described by Jaglic et al. (14).

Thirty-day-old rabbits were divided into groups of eight rabbits (four males and four females) in each group. Each group was housed in an isolated room, and two rabbits (one male and one female) from the same group were kept in a cage. Rabbits from the subcutaneous group and the intranasal group were subcutaneously and intranasally inoculated with $6.0 \times 10^4$ CFU of the isolate suspended in 100 µL of sterile normal saline, respectively. Rabbits from the intranasal control group and subcutaneous control group were intranasally and subcutaneously inoculated with 100 µL of sterile normal saline, respectively. The bacterial concentration and infection routes of the rabbit infection experiment were selected according to the previous reports of Jaglic et al. (14) and Wang et al., (16).

Twenty-day-old white feather broilers were divided into groups of eight broilers in each group. Broilers from the same group were kept in a cage and housed in a separate room. Broilers from the intratracheal group were intratracheally inoculated with $6.0 \times 10^4$ CFU of the isolate suspended in 100 µL of sterile normal saline. Broilers from the intratracheal control group were intratracheally inoculated with 100 µL of sterile normal saline. The infection route and bacterial concentration of the broiler infection experiment were selected as described in the previous reports of Peng et al. (21) and Jaglic et al. (22), respectively.

The rabbits and broilers had free access to feed and water during the 15-day experiment period, and the rabbits and broilers were monitored for clinical signs of infection during the entire experiment period. All of the rabbits and the broilers were examined for the presence of pathological lesions at the end of the experiment. Tracheas, lungs, livers, hearts, spleens, kidneys, and whole blood of all rabbits and broilers as well as the bursa of Fabricius of all broilers were collected for bacteriological

examination using the multiplex PCR assays developed by Townsend et al. (3) and Harper et al. (18). The lung samples of rabbits from intranasal group and intranasal control group as well as the lung samples from the all broilers were used for the histological examination using hematoxylin-eosin staining.

Before inoculation, rabbits (40 mg/kg) and broilers (20 mg/kg) were anesthetized by intravenous injection of ketamine to prevent or minimize the suffering. The inoculated rabbits and broilers that were caught up in the endpoint of the disease (dyspnea, loss of appetite, or weight loss of 15% or more) were euthanized by bleeding from the jugular vein under ketamine narcosis. At the end of the experiment, the survived rabbits and broilers were also sacrificed by bleeding from the jugular vein under ketamine narcosis.

## Genome sequencing and comparative analyses

The genomes of the isolates were determined using Oxford Nanopore Technologies (ONT) at Tianjin Novogene Bioinformatic Technology Co., Ltd. (Tianjin, China). The total reads generated by the ONT ranged from 55,097 to 403,740 for the 19 isolates, and the total base ranged from 509,568,174 to 2,213,944,017, with the $N_{50}$ value ranged from 9,701 to 21,834 and the genome coverage ranged from 207- to 887-fold. The raw reads were analyzed using the NanoPlot (version 1.30.1) and filtered using the NanoFilt (version2.7.1) (the parameters –*headcrop* and –*tailcrop* of 10 were applied), and then the filtered reads were *de novo* assembled using the Unicycler (version 0.4.8). The annotation of the genome was performed using the NCBI Prokaryotic Genome Annotation Pipeline. The prophage and genomic island in the genome were determined using the web servers of PHAST (http://phast.wishartlab.com/) and IslandViewer 4 (https://www.patho-genomics.sfu.ca/islandviewer/), respectively. To perform the comparative analyses, whole genomic sequences of *P. multocida* serogroup A strains 36950 (CP003022), ATCC 43137 (CP008918), and CQ2 (CP033599); serogroup B strains Ban-PM7 (CP052765), sample-B (CP133836), and Tibet-Pm1 (CP072655); serogroup D strains 102426 (CP135194), HN06 (CP003313), and P504190 (CP110621); and serogroup F strains 9N (CP028927), 36502 (CP097792), Pm70 (AE004439), HN07 (CP007040), s4 (CP084165), CIRMBP-0873 (CP020347), CIRMBP-0884 (CP020345), and SD11 (CP090520) were freely obtained from NCBI database. Comparative genome analyses were performed using the Mauve (version 2.4.0) (23), BLAST (version 2.4.0) (24), BRIG (version 0.95-dist) (25) and Easyfig (version 2.2.5) (26).

## Statistical analysis

All data were analyzed using Microsoft Office Excel 2010 (Microsoft Corporation, the USA), and then the statistical summaries were generated. The *t*-test was performed to evaluate the differences between the infected rabbits and the control rabbits, and *P*-value less than 0.05 was considered to be statistically significant.

## RESULTS

### Capsular and LPS genotyping

The capsular and LPS genotyping multiplex PCR assays produced segments of approximately 850 bp and 470 bp in length for the 19 isolates, respectively. The sequences of segments from the capsular and LPS genotyping multiplex PCR assays shared the highest identity (up to 99.90% identity) with the *fcbD* and *gatF* of *P. multocida*, respectively. The results suggested that the serotypes of the 19 isolates were F:L3 (Table S1).

### Multi-locus sequence typing

The 19 isolates were typed into 4 sequence types using the multi-host MLST scheme. ST191 was the most dominant sequence type (9/19), followed by ST12 (5/19), ST193 (3/19), and ST192 (2/19) (Table S1). The 19 isolates were grouped into three clusters based on the concatenated housekeeping genes of the multi-host MLST scheme. Isolates

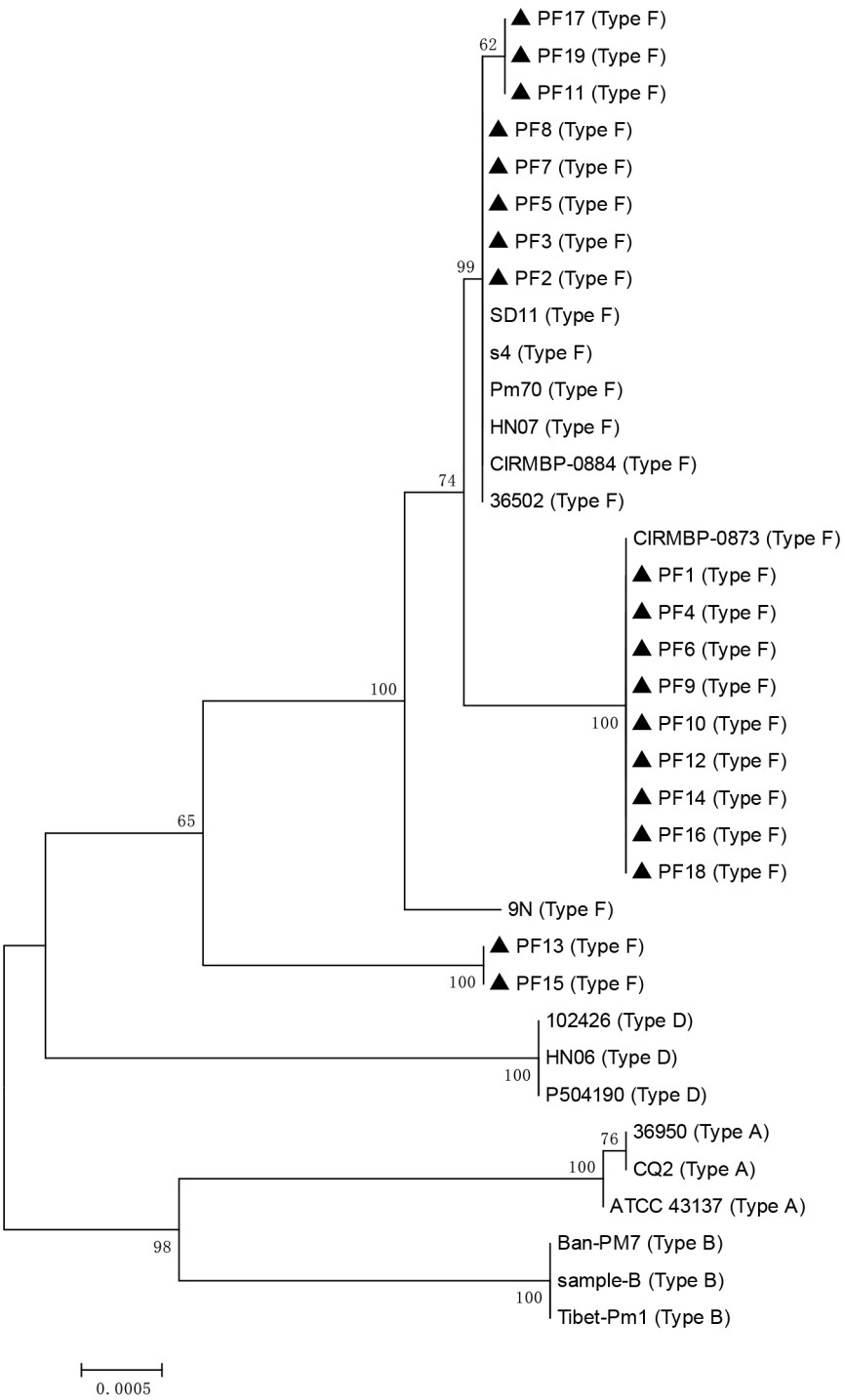

**FIG 1** Neighbor-joining tree indicating the positions of the 19 rabbit-sourced *P. multocida* isolates (PF1–PF19) based on the housekeeping genes of multi-host MLST and RIRDC MLST schemes. The multi-host MLST and RIRDC MLST housekeeping genes of the isolate were concatenated, and then the neighbor-joining phylogenetic tree (1,000 bootstrap replications) was constructed using MEGA 5.0. The multi-host MLST and RIRDC MLST housekeeping genes of the strains were freely obtained from the NCBI database under the accession numbers: Pm70 (AE004439), HN07 (CP007040), s4 (CP084165), CIRMBP-0873 (CP020347), CIRMBP-0884 (CP020345), 9N (CP028927), AH09 (CP090521), SD11 (CP090520), and 36502 (CP097792).

including PF2, PF3, PF5, PF7, PF8, PF11, PF17, and PF19 were relative to SD11 (type F, rabbit), s4 (type F, rabbit), Pm70 (type F, chicken), HN07 (type F, pig), CIRMBP-0884 (type F, rabbit) and 36502 (type F, chicken), isolates including PF1, PF4, PF6, PF9, PF10, PF12, PF14, PF16, and PF18 were relative to CIRMBP-0873 (type F, rabbit), while isolates PF13 and PF15 were grouped into a cluster without any other *P. multocida* serogroup F strains (Fig. 1).

## Virulence genes determination

The virulence gene profiles of the 19 isolates were identical. The virulence genes of *fimA*, *hsf-1*, *hsf-2*, *pfhB2*, *ptfA*, *tadD*, *exbB*, *exbD*, *fur*, *hgbA*, *hgbB*, *tonB*, *nanB*, *nanH*, *ompA*, *ompH1*, *ompH2*, *ompW*, *oma87*, *sodA,* and *sodC* were positive for the 19 isolates, whereas the virulence genes of *pfhA*, *pfhB1*, *toxA*, *tbpA,* and *pmHAS* were negative for the 19 isolates.

## Antimicrobial susceptibility test

The results of the antimicrobial susceptibility test showed that all of the 19 *P. multocida* serogroup F isolates were sensitive to ceftriaxone, cefotaxime, ciprofloxacin, ofloxacin, and rifampin, and all the 19 isolates were resistant to ampicillin. Among the 19 isolates, 12 (12/19, 63.16%), 12 (12/19, 63.16%), 8(8/19, 42.11%), and 3 (3/19, 15.79%) isolates were resistant to clarithromycin, chloramphenicol, telithromycin, and tetracycline, respectively.

## Animal experiments

Before inoculation, all of the rabbits and broilers were confirmed to be free of *P. multocida* and negative for IgG antibodies against *P. multocida*. Rabbits and broilers from control groups remained free of *P. multocida* and negative for *P. multocida* IgG at the end of the experiment, and no clinical signs and gross lesions were observed in these animals.

All 19 isolates caused severe disease in the subcutaneously inoculated rabbits, and most of the inoculated rabbits [ranged from 50% (4/8) to 87.5% (7/8)] were euthanized during the 15-day experiment period (Table S2). Rabbits that were euthanized within 24 h post-inoculation became seriously ill, and diffuse hemorrhagic pneumonia

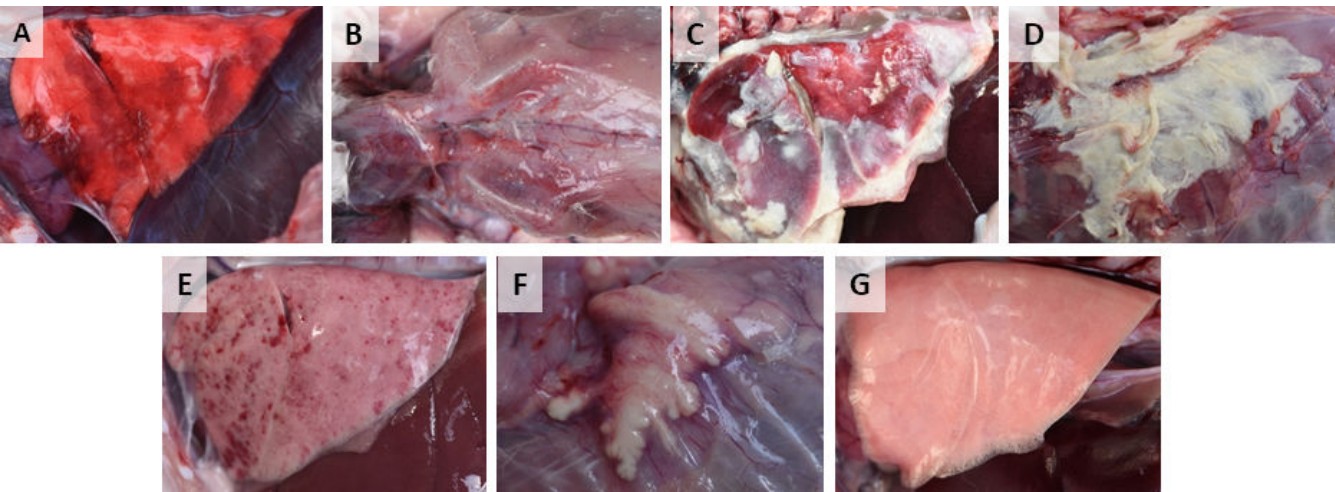

**FIG 2** Pathologic lesions of the rabbits subcutaneously inoculated with the 19 isolates. Rabbits were inoculated with $6.0 \times 10^4$ CFU of the isolate. Diffuse hemorrhagic pneumonia (A) and diffuse subcutaneous hemorrhage (B) were observed in the rabbits that were euthanized within 24 h post-inoculation. Fibrino-hemorrhagic pleuropneumonia (C) and diffuse subcutaneous abscess (D) were observed in the rabbits that were euthanized after 24 h post-inoculation. Hemorrhagic pneumonia (E) and local subcutaneous abscess (F) were observed in the rabbits that survived the experiment. No pathologic lesions were observed in the lungs of the subcutaneous control rabbits (G).

(Fig. 2A) and diffuse subcutaneous hemorrhage around the inoculation site (Fig. 2B) were observed in these rabbits. Rabbits that were euthanized after 24 h post-inoculation showed fibrino-hemorrhagic pleuropneumonia (Fig. 2C) and diffuse subcutaneous abscess around the inoculation site (Fig. 2D). Rabbits that survived the experiment showed hemorrhagic pneumonia (Fig. 2E) and local subcutaneous abscess at the inoculation site (Fig. 2F). All of the 19 isolates could be detected in tracheas, lungs, and livers from the subcutaneously inoculated rabbits, and the presence of the isolates in hearts (12/19), spleens (15/19), kidneys (14/19), and whole blood (12/19) from the subcutaneously inoculated rabbits could also be detected. Interestingly, rabbits that survived the subcutaneous inoculation were positive for *P. multocida* IgG at the end of the experiment.

The intranasal inoculation of all 19 isolates caused severe disease in the inoculated rabbits but no acute disease in these rabbits. Clinical signs including cough, nasal discharge, and dyspnea could be observed in the inoculated rabbits, and a high proportion of inoculated rabbits [ranged from 37.5% (3/8) to 75% (6/8)] were euthanized during the 15-day experiment period (Table S2). Rabbits that were euthanized showed fibrino pleuropneumonia (Fig. 3A) or fibrinopurulent pleuropneumonia (Fig. 3B), whereas rabbits that survived the experiment showed pulmonary consolidation with hemorrhagic pneumonia (Fig. 3C). Inflammatory exudates in bronchiole and alveoli, penetration of red blood cells in bronchiole and alveoli as well as degeneration of the alveolar epithelial cells were observed in the lungs showing fibrino pleuropneumonia or fibrinopurulent pleuropneumonia (Fig. 3E). Inflammatory exudates in bronchiole and alveoli and proliferation of alveolar epithelial cells were observed in the lungs showing pulmonary consolidation with hemorrhagic pneumonia (Fig. 3F). All of the 19 isolates could be detected in the tracheas, lungs, and livers from the intranasally inoculated rabbits, and the isolates could also be detected in hearts (9/19), spleens (13/19), kidneys (8/19), and whole blood (10/19) from the intranasally inoculated rabbits. Moreover, rabbits that survived the intranasal inoculation were positive for IgG against *P. multocida*.

All of the intratracheally inoculated broilers survived the challenge showing no clinical signs of the infection. At the end of the experiment, no gross lesions were observed in the inoculated broilers (Fig. S1). The tissue samples including tracheas, lungs,

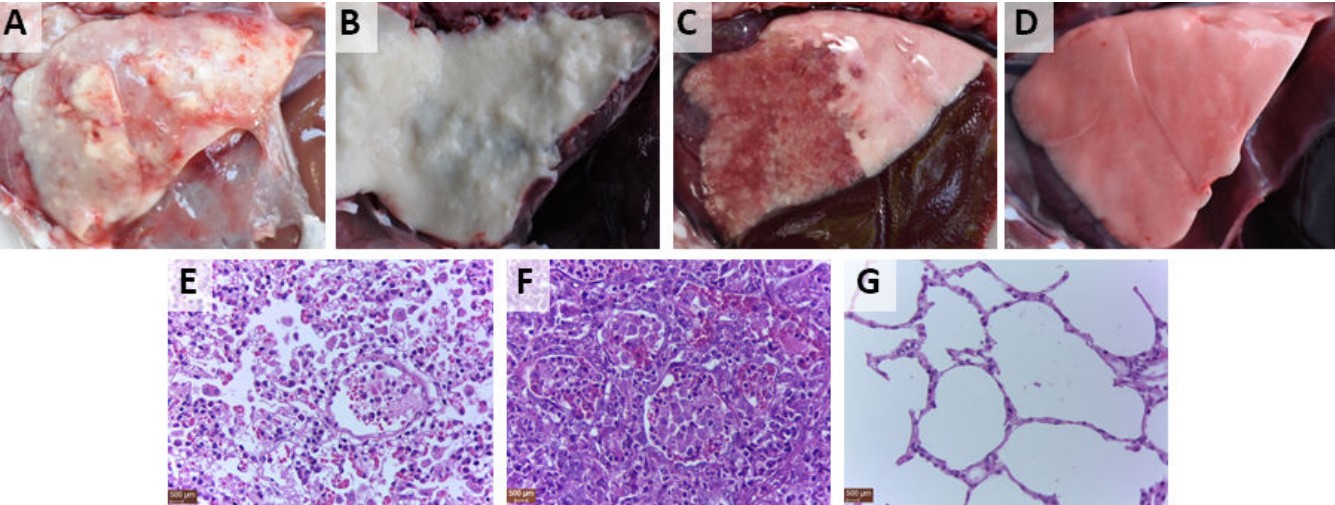

**FIG 3** Pathologic and histological lesions of the rabbits intranasally inoculated with the 19 isolates. Rabbits were inoculated with $6.0 \times 10^4$ CFU of the isolate. Fibrino pleuropneumonia (A) or fibrinopurulent pleuropneumonia (B) was observed in the rabbits that were euthanized during the 15-day experiment period. Pulmonary consolidation with hemorrhagic pneumonia (C) was observed in the rabbits that survived the experiment. No pathologic lesions were observed in the lungs of the intranasal control rabbits (D). Inflammatory exudates in the bronchiole and alveoli, penetration of red blood cells in the bronchiole and alveoli as well as degeneration of the alveolar epithelial cells (E) were observed in the lungs with fibrino pleuropneumonia or fibrinopurulent pleuropneumonia. Inflammatory exudates in bronchiole and alveoli and proliferation of alveolar epithelial cells (F) were observed in the lungs with pulmonary consolidation and hemorrhagic pneumonia. No histopathological lesions (G) were observed in the lungs of the intranasal control rabbits.

livers, hearts, spleens, kidneys, blood, and bursa of Fabricius from inoculated broilers were negative for *P. multocida*, the *P. multocida* IgG was also negative for these broilers, and no histopathological changes were observed in the lungs of the inoculated broilers (Fig. S1).

## Genome sequencing and comparative analyses

The genome sizes of the 19 isolates ranged from approximately 2.46 to 2.49 Mbp, with the average G + C content between 40.3% and 40.35% (Table S3). The predicted genes of the 19 isolates ranged from 2409 to 2473, with the protein-coding sequences between 2288 and 2356 (Table S3). All of the 19 isolates contained prophages and genomic islands, and most of the genomic islands were located in the prophage regions (Table S4).

The genomes of 17 isolates including PF2, PF3, PF4, PF5, PF6, PF7, PF8, PF9, PF10, PF11, PF12, PF14, PF15, PF16, PF17, PF18, and PF19 were composed of a single chromosome, whereas the genomes of the PF1 and PF13 were composed of a chromosome and a plasmid (Table S4). The plasmid of the PF1 was 53,946 bp in length, in which an intact prophage of 53,137 bp in length was determined (Fig. S2). The plasmid of the PF13 was 260,620 bp in length, in which an intact prophage (37,778 bp in length) and an incomplete prophage (10,700 bp in length) were determined (Fig. S2). Interestingly, except for the intact and incomplete prophage sequences in the plasmid of PF13, the remaining sequences of PF13 plasmid were highly homologous (up to 100% identity) with the genomic sequences of other *P. multocida* strains (Fig. 4).

To understand why the 19 isolates were highly pathogenic for rabbits, comparative genomic analyses were performed between the 19 isolates and the rabbit-sourced low-virulent *P. multocida* serogroup F strain s4. The genomic sizes of the 19 isolates (ranging from 2.46 to 2.49 Mbp) were larger than that of s4 (the genomic size of s4 is approximately 2.06 Mbp). The genomes of the 19 isolates showed highly collinear with that of s4, and three collinear patterns were determined between the genomes of the 19 isolates and that of s4 (Fig. S3), which corresponded to the phylogenetic analysis result based on the housekeeping genes of the multi-host MLST scheme (Fig. 1). Furthermore, specific sequences were determined in the genomes of the 19 isolates by comparison with that of s4. About 200 functional genes were determined in the specific sequences, and these functional genes were involved in many vital bacterial physiological processes such as genetic information processing, environmental information processing, lipid metabolism, amino acid metabolism, and energy metabolism (Fig. S4), and these functional genes might contribute to the fitness and invasion of the 19 isolates in rabbits.

All of the 19 rabbit-sourced high-virulent isolates were avirulent to white feather broilers. Therefore, the genomes of the 19 isolates were compared with that of Pm70 a well-characterized avian-sourced *P. multocida* serogroup F strain recovered from chickens with fowl cholera (27). Interestingly, three virulence genes *pfhB1* (PM0057), *lspB_2* (PM0058), and *pfhB2* (PM0059) were identified in the Pm70 but absent in the 19 isolates. In addition, five virulence genes (VM82_04605, VM82_06595, VM82_07500, VM82_10495, and VM82_10500) (21, 28) that contributed to fowl cholera were also absent in the 19 isolates.

The entire capsule biosynthesis loci of the 19 isolates showed significant similarity (ranged from 99.75% to 99.95%) to that of the *P. multocida* serogroup F strain Pm70, and the entire capsule biosynthesis loci of the 19 isolates showed a high level of similarity (ranged from 95.76% to 95.81%) to that of the *P. multocida* serogroup A strain CQ2. However, the similarities between the entire capsule biosynthesis loci of the 19 isolates and that of the *P. multocida* serogroup D strain HN06 only ranged from 64.20% to 64.23% (Fig. 5).

All of the 19 isolates possessed 18 LPS biosynthetic genes and produced the LPS of genotype L3. All of the 19 isolates produced the LPS with two inner core structures (glycoforms A and B) because of the presence of the biosynthesis genes *kdtA* and *kdkA*

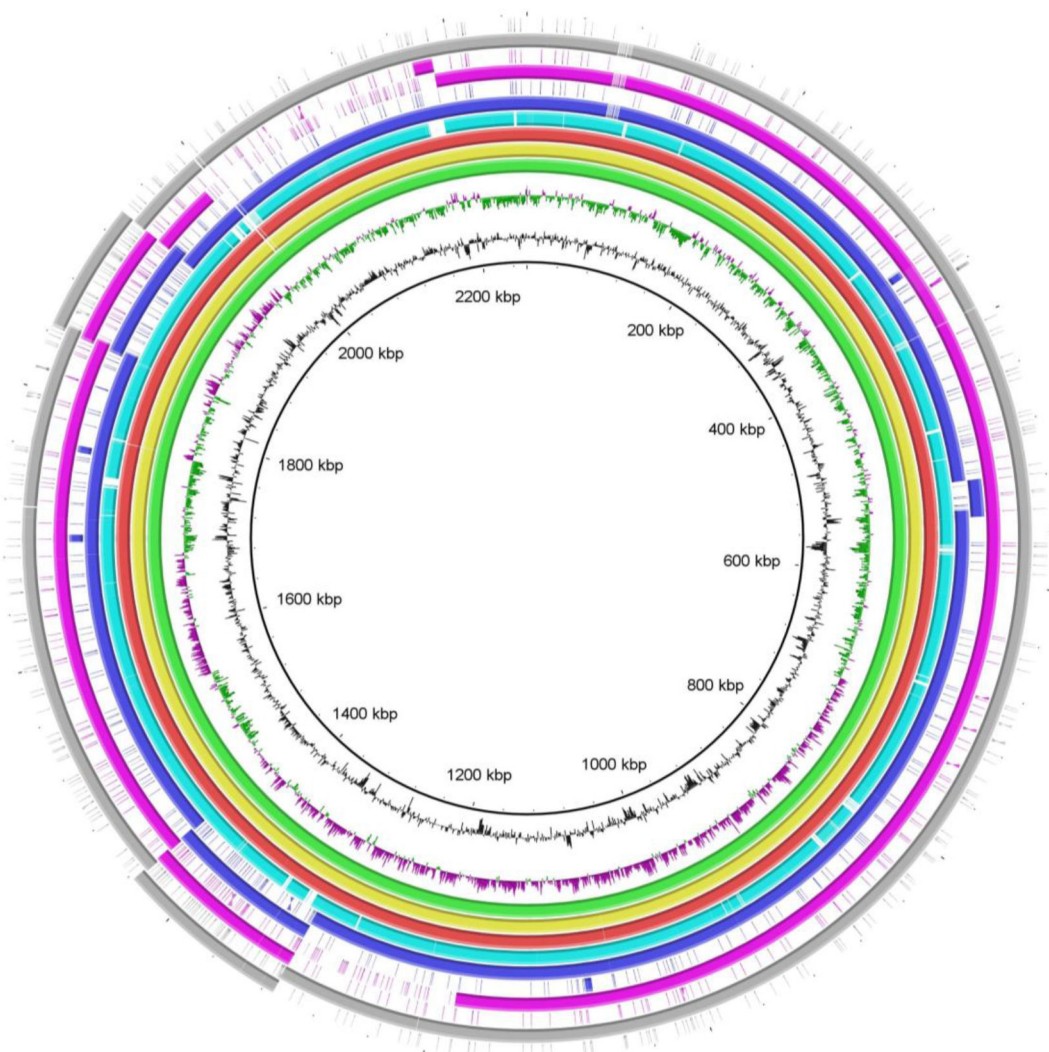

**FIG 4** Comparison between the genome of PF13 and those of s4, CIRMBP-0884, HN07, HN06, CQ2, and Pm70. From the outside to the inside, circle 1 (gray): plasmid of PF13; circle 2 (gray): chromosome of PF13; circle 3 (purple): plasmid of s4; circle 4 (purple): chromosome of s4; circle 5 (blue): plasmid of CIRMBP-0884; circle 6 (blue): chromosome of CIRMBP-0884; circle 7 (cyan-blue): HN07; circle 8 (red): HN06; circle 9 (yellow): CQ2; circle 10 (green): Pm70; circles 11 and 12 represent the G + C content and GC skew, respectively; the innermost circle represents DNA base position.

(29). All of the 19 isolates might produce the LPS with the full-length outer core structure because of the presence of the LPS outer core biosynthetic genes *natC*, *gatG*, *natB*, *gatF*, *gctC,* and *hptE* (29). Interestingly, a 183 bp nucleotide sequence in-frame deletion (in positions 559–741) in the LPS outer core biosynthetic gene *natC* and a 210 bp nucleotide sequence N-terminal redundancy (in positions 1–210) in the LPS outer core biosynthetic gene *gatF* were determined in the 19 isolates by comparison with the Pm70 (Fig. 6). The LPS outer core biosynthetic genes including *gatG*, *natB*, *gctC,* and *hptE* showed significant similarity among the *P. multocida* serotype F:L3 strains deposited in the NCBI nucleotide database, whereas polymorphisms were determined in the LPS outer core biosynthetic genes *natC* and *gatF* among these serotype F:L3 strains (Fig. 6).

Pm70 (chicken) and 9N (rodent) produce the full-length glycosyltransferase NatC of 392 amino acids in length. The rabbit-sourced strains including PF1–PF19, s4, CIRMBP-0873, CIRMBP-0884, and AH09 produce the truncated NatC of 331 amino acids in length because of the in-frame 61 amino acids deletion in positions 183 to 243. SD11 (rabbit), HN07 (swine), F (bovine), and 36502 (chicken) produce two hypothetical proteins that were fully matched with the N-terminal (in positions 1–136) and C-terminal

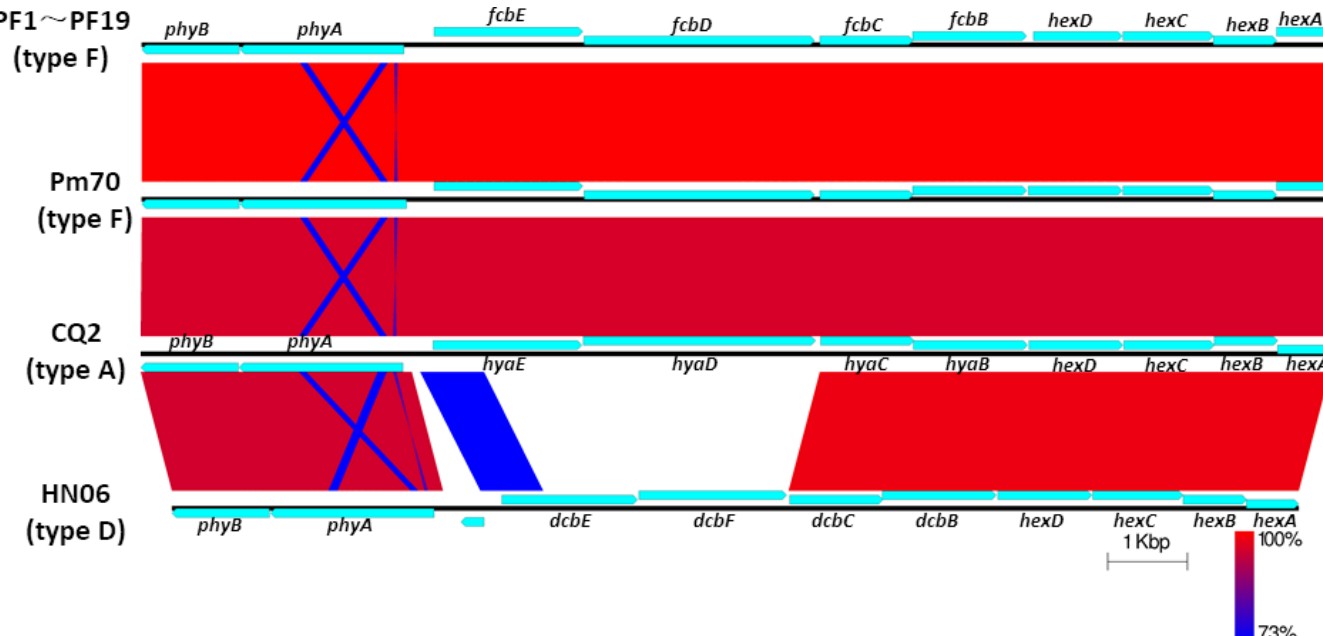

**FIG 5** Comparative analyses of the entire cap locus between the 19 isolates and other *P. mltocida* strains. The entire cap locus of the 19 isolates was compared with those of Pm70 (type F), CQ2 (type A), and HN06 (type D). The color code represents the BLASTn identity.

(in positions 171–392) of the NatC of Pm70 because of the G410A mutation in the sequence of *natC* (Fig. 7A).

Pm70 (chicken) produces the shortest glycosyltransferase GatF of 208 amino acids in length. F (bovine) produces the GatF of 260 amino acids in length with 52 amino acids redundancy at the N-terminal by comparison with that of Pm70. PF1–PF19 (rabbit), s4 (rabbit), AH09 (rabbit), SD11 (rabbit), and 36502 (chicken) produce the GatF of 278 amino acids in length with 70 amino acids redundancy at the N-terminal by comparison with that of Pm70. CIRMBP-0873 (rabbit), HN07 (swine), and 9N (rodent) produce the longest GatF of 280 amino acids in length with 72 amino acids redundancy at the N-terminal by comparison with that of Pm70. CIRMBP-0884 (rabbit) produces the GatF of 268 amino acids in length with 70 amino acids redundancy at the N-terminal and a truncated C-terminal of MQK due to frameshift mutation (the C-terminal of GatF of the other *P. multocida* serotype F:L3 strains were NAKMKLKCIVKFE) by comparison with that of Pm70 (Fig. 7B).

## DISCUSSION

Previous studies showed that *P. multocida* serogroup F was involved in the respiratory disease of rabbits (11, 12), and *P. multocida* serogroup F isolates of different sequence types (11, 12) and pulsed-field gel electrophoresis types (30) were recovered from rabbits. In accordance with these results, the 19 *P. multocida* serogroup F isolates in the present study were typed into four sequence types using a multi-host MLST scheme. Moreover, phylogenetic analyses showed that all of the *P. multocida* serogroup F strains (isolated from swine, bovine, avian, rabbits, and rodents) deposited in the NCBI database were grouped into three distinct clusters. Taken together, these findings suggest the genetic diversity of *P. multocida* serogroup F.

It was shown that *P. multocida* serogroup F was pathogenic for rabbits (14, 16, 22). The rabbit-sourced *P. multocida* serogroup F strain J-4103 caused severe gross lesions and high mortality in rabbits (14). The avian-sourced *P. multocida* serogroup F strains P-4218 (turkey) and C21724H3km7 (chicken) could also cause severe diseases and significant mortality in rabbits (22). However, our previous study showed that the rabbit-sourced *P. multocida* serogroup F strain s4 was a low-virulent strain, which caused mild patho-

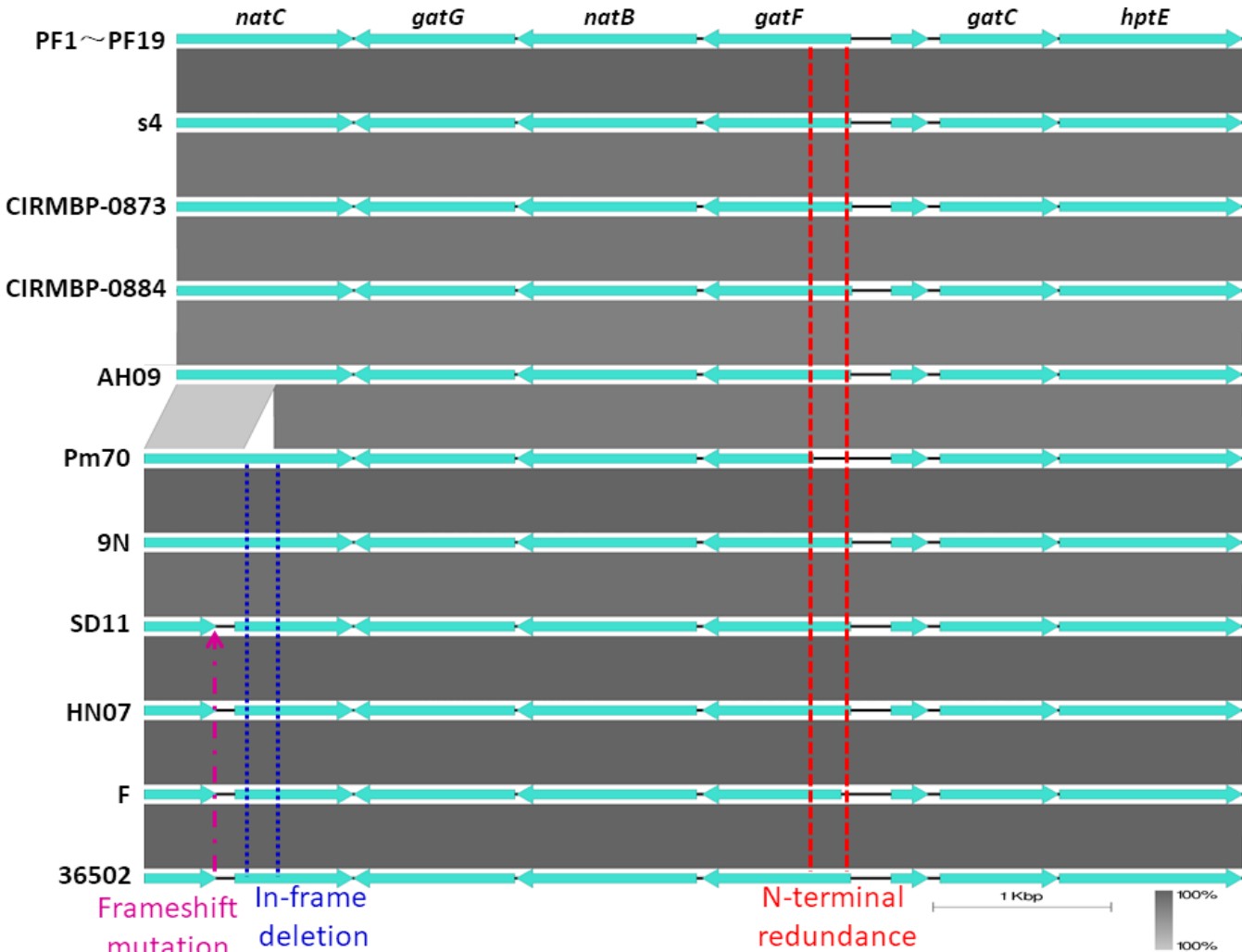

**FIG 6** Comparative analyses of the LPS outer core locus between the 19 isolates and other *P. multocida* serogroup F:L3 strains. The LPS outer core biosynthetic genes of the 19 isolates were compared with those of s4, CIRMBP-0873, CIRMBP-0884, AH09, Pm70, 9N, SD11, HN07, F, and 36502.

genic lesions in most of the challenged rabbits (16). In this study, all of the 19 isolates were highly pathogenic for rabbits because all of the isolates could result in severe gross lesions and significant mortality in inoculated rabbits as the previously described high-virulent strains J-4103 (14), P-4218 (22), and C21724H3km7 (22) did. Notably, the infection of J-4103 (14), P-4218 (22), and C21724H3km7 (22) could result in septicemic diseases in rabbits, whereas the 19 isolates in this study and s4 (16) caused purulent disease. The results would be helpful for understanding the pathogenic variation of *P. multocida* serogroup F.

*P. multocida* is an important pathogen that infects a wide range of animals (1, 2). However, the study of Peng et al. showed that the swine-sourced high-virulent *P. multocida* serogroup F strain HN07 that could kill pigs was avirulent to chickens (21), suggesting the infection of *P. multocida* displays host specificity. In the present study, all of the 19 rabbit-sourced high-virulent isolates were also avirulent to white feather broilers. Several studies have tried to explore the molecular basis for host specificity of *P. multocida* using comparative genomics (20, 21). Peng et al. identified a number of virulence genes in the genomes of avian virulent strains (P1059, X73, and GX) but absent in that of swine-sourced strain HN07 (21), which might contribute to the avirulence of HN07 in chickens. By comparing the genomes of 13 bovine hemorrhagic septicemia (HS) strains with those of non-HS strains 36950 (bovine), 3480 (swine), HN06 (swine), and

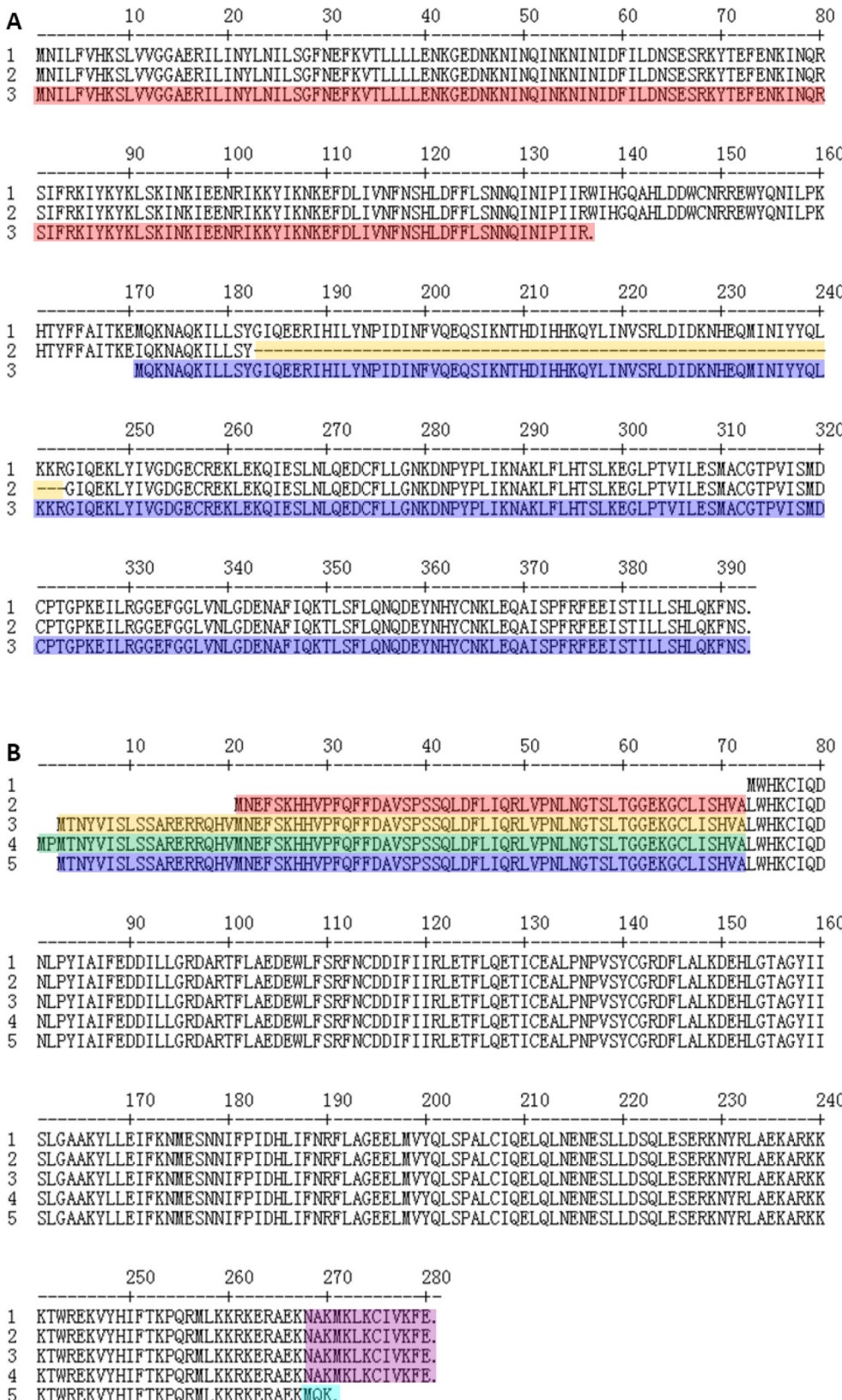

FIG 7 Comparative analyses of the LPS outer core biosynthetic glycosyltransferases NatC and GatF between the 19 isolates and other *P. multocida* serogroup F:L3 strains. (A) Pm70 and 9N produce the full-length NatC of 392 amino acids in length (1). PF1–PF19, s4, CIRMBP-0873, CIRMBP-0884, and AH09 produce the truncated NatC of 331 amino acids in length because

**FIG 7** (Continued)

of the in-frame 61 amino acid truncation (yellow) (2). SD11, HN07, F, and 36502 produce two hypothetical proteins that were fully matched with the N-terminal (red) and C-terminal (blue) of the NatC of Pm70 (3). (B) Pm70 produces the shortest GatF of 208 amino acids in length (1). F produces the GatF of 260 amino acids in length with 52 amino acids redundancy at the N-terminal by comparison with that of Pm70 (red) (2). PF1–PF19, s4, AH09, SD11, and 36502 produce the GatF of 278 amino acids in length with 70 amino acids redundancy at the N-terminal by comparison with that of Pm70 (yellow) (3). CIRMBP-0873, HN07, and 9N produce the longest GatF of 280 amino acids in length with 72 amino acids redundancy at the N-terminal by comparison with that of Pm70 (green) (4). CIRMBP-0884 (rabbit) produced the GatF of 268 amino acids in length with 70 amino acids redundancy at the N-terminal (blue) and a truncated C-terminal of MQK (cyan-blue) [the GatF C-terminal of the other *P. multocida* serogroup F strains were NAKMKLKCIVKFE (purple)] by comparison with that of Pm70 (5).

Pm70 (chicken), Moustafa et al. identified a set of 96 genes unique to the bovine HS strains (20). In the present study, several virulence genes that contribute to fowl cholera were absent in the 19 isolates. Furthermore, by comparison with the LPS biosynthetic genes of *P. multocida* serotype F:L3 strains, deletions and redundancies were determined in the genes of *natC* and *gatF*, respectively. Interestingly, the sequences of *natC* and *gatF* from rabbit-sourced strains were identical, suggesting that the *natC* and *gatF* of the rabbit-sourced strains might be selected by the rabbit immune pressure and associated with the host specificity. In contrast to these results, a larger-scale comparative genomic analysis based on 109 *P. multocida* strains of different hosts with multiple types of diseases showed that there were no genes of *P. multocida* that were specific to a particular type of host (31). Taken together, the knowledge of the host specificity of *P. multocida* is still limited and controversial, and the molecular basis for the host specificity of the pathogen needs to be investigated further.

In conclusion, the pathogenicity and genomic features of 19 rabbit-sourced *P. multocida* serogroup F isolates were determined in the present study. Our findings showed the genetic diversity and pathogenic variation of *P. multocida* serogroup F. Moreover, the results of the present study would also be helpful for the understanding of the host specificity of *P. multocida*.

## ACKNOWLEDGMENTS

This work was supported by the Fujian Public Welfare Project (2022R10260012), the Construction of Science and Technology Innovation Team of Fujian Academy of Agricultural Sciences (CXTD2021007-2), 5511 Collaborative Innovation Project of Fujian Academy of Agricultural Sciences (XTCXGC2021008), and the Earmarked Fund for China Agriculture Research System (CARS-43-G-5).

W.J. and X.X. conceived the study. W.J., S.S., C.D., G.C., and S.L. performed the experiments. W.J. drafted the manuscript. All authors read and approved the final manuscript.

## AUTHOR AFFILIATIONS

[1]Institute of Animal Husbandry and Veterinary Medicine, Fujian Academy of Agricultural Sciences, Fuzhou, Fujian, China
[2]Fujian Key Laboratory of Animal Genetics and Breeding, Fuzhou, Fujian, China

## AUTHOR ORCIDs

Jinxiang Wang ⓘ http://orcid.org/0000-0001-8911-3068

## FUNDING

| Funder | Grant(s) | Author(s) |
| --- | --- | --- |
| Fujian Public Welfare Project | 2022R10260012 | Jinxiang Wang |

| Funder | Grant(s) | Author(s) |
|---|---|---|
| Construction of Science and Technology Innovation Team of FAAS | CXTD2021007-2 | Jinxiang Wang |
| 5511 Collaborative Innovation Project of FAAS | XTCXGC2021008 | Jinxiang Wang |
| Earmarked Fund for China Agriculture Research System | CARS-43-G-5 | Xiping Xie |

## DATA AVAILABILITY

The complete genome sequences of the 19 isolates used in this study were deposited in the NCBI GenBank (https://www.ncbi.nlm.nih.gov/genbank/): PF1 (CP112898, CP112899), PF2 (CP111081), PF3 (CP111082), PF4 (CP111083), PF5 (CP111142), PF6 (CP111143), PF7 (CP111144), PF8 (CP113236), PF9 (CP111145), PF10 (CP111146), PF11 (CP111147), PF12 (CP112891), PF13 (CP113522, CP113523), PF14 (CP112892), PF15 (CP112893), PF16 (CP112894), PF17 (CP112895), PF18 (CP112896), and PF19 (CP112897). The data sets used and/or analyzed during the current study are available from the corresponding author upon reasonable request.

## ETHICS APPROVAL

The present study was approved by the Research Ethics Committee of the Institute of Animal Husbandry and Veterinary Medicine, Fujian Academy of Agricultural Sciences (FAAS). The approval numbers were FAAS-AHVM2022-0303R and FAAS-AHVM2022-0303C for the experimental infections of rabbits and white feather broilers, respectively. All the animal experiments were carried out according to the Guidelines for the operation of laboratory animals issued by the Institute of Animal Husbandry and Veterinary Medicine, FAAS.

## ADDITIONAL FILES

The following material is available online.

### Supplemental Material

**Figure S1 (Spectrum03654-23-s0001.pdf).** Pathologic and histopathological lesions in the white feather broilers.
**Figure S2 (Spectrum03654-23-s0002.pdf).** Circular maps of the plasmid of PF1 and PF13.
**Figure S3 (Spectrum03654-23-s0003.pdf).** Colinear analyses of the genomes between the 19 isolates and that of s4.
**Figure S4 (Spectrum03654-23-s0004.pdf).** Functional genes identified in the genomes of the 19 isolates but absent in that of s4.
**Table S1 (Spectrum03654-23-s0005.docx).** Details of the isolates.
**Table S2 (Spectrum03654-23-s0006.docx).** Numbers of rabbits euthanized and survived.
**Table S3 (Spectrum03654-23-s0007.docx).** Genome overview of the isolates.
**Table S4 (Spectrum03654-23-s0008.docx).** Prophage sequences and gnomic islands of the isolates.

### Open Peer Review

**PEER REVIEW HISTORY (review-history.pdf).** An accounting of the reviewer comments and feedback.

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
