## [Reviewer comments · Microbiology Spectrum]

Microbiology Spectrum

Pathogenic and genomic characterisation of rabbit sourced *Pasteurella multocida* serogroup F isolates recovered from dead rabbits with respiratory disease

Jinxiang Wang, Shikun Sun, Dongjin Chen, Chenfang Gao, Lei Sang, and Xiping Xie

Corresponding Author(s): Jinxiang Wang, Institute of Animal Husbandry and Veterinary Medicine, Fujian Academy of Agricultural Sciences

Review Timeline:

Submission Date:	October 12, 2023
Editorial Decision:	January 2, 2024
Revision Received:	January 23, 2024
Editorial Decision:	January 24, 2024
Revision Received:	January 25, 2024
Accepted:	January 26, 2024

Editor: Artem Rogovskyy

Reviewer(s): Disclosure of reviewer identity is with reference to reviewer comments included in decision letter(s). The following individuals involved in review of your submission have agreed to reveal their identity: Zhong Peng (Reviewer #1)

Transaction Report:

DOI: <https://doi.org/10.1128/spectrum.03654-23>

Re: Spectrum03654-23 (Pathogenic and genomic characterisation of rabbit sourced *Pasteurella multocida* serogroup F isolates recovered from dead rabbits with respiratory disease)

Dear Dr. Jinxiang Wang:

Thank you for the privilege of reviewing your work. Below you will find my comments, instructions from the Spectrum editorial office, and the reviewer comments.

Revision Guidelines

Sincerely,
Artem Rogovskyy
Editor
Microbiology Spectrum

Reviewer #1 (Comments for the Author):

In this study, the authors reported the virulence and complete genome sequencing of nine rabbit originated *Pasteurella multocida* serogroup F. The authors assessed the virulence of these nine strains in both rabbit and chicken models and generated the complete genome sequences using Nanopore technology. The authors also attempted to figure out pathogenesis related genes by performing comparative genomics. However, this study also displayed several points of weakness.

1. One of my great concern is the novelty. What is the difference between this study and previously studies assessing the pathogenicity of rabbit-originating serogroup F strains using the same models (references 11, 12, 14, 16 as the authors cited in the manuscript)
 2. Fig. 1, the authors only performed phylogenetic analysis based on the MLST data, why not generate a tree using the whole genome sequences as those data have been generated using Nanopore sequencing? In addition, strains belonging to the other serogroups should be also included in the MLST tree.
 3. Line 109, why both multihost and RIRDC typing methods were used? In the current PubMLST online version, only multihost MLST is recommended.
 4. Line 119, you have obtained the complete genome sequences, why not analyze more VFGs?
 5. Line 171, more information about the sequencing should be added. For example, how many raw data are generated? how did the low-quality data be removed? what is the criterion for defining the low-quality data? how did the phylogenetic trees be generated? etc.
 6. Line 256, it would be surprising that no-IgG was detected. Even the strains were avirulent to chickens, antibodies should be induced.
 7. Line 272, have you tested the corresponding phenotypes?
 8. Animal tests, both high-dose and low-dose challenged groups should be considered as you did not know the virulence of these strains to rabbits before study. Why only chose this dose for inoculation?
 9. Lines 297-299, while comparative genomic analysis being performed, no useful information is available. Why these nine strains were virulent to rabbits? Did they carry any virulence associated genes that were absent from the avirulent strain S4?
 10. Line 300, remove unexpectedly. It has been well known that *P. multocida* strains recovered from non-avian species generally do not cause fowl cholera.
 11. Lines 320-324, several deletions or redundant bases were identified. Did these deletions and redundant bases display any impacts on the biosynthesis or completeness of the LPSs of these nine serogroup F strains ?
 12. Lines 392-348, did these differences affect the virulence of the nine serogroup F strains ?
 13. Line 351, I do not think serogroup F is "typical avian-adapt" as this serogroup has been widely characterized in different host species.
 14. Line 392, I have read these two articles (reference 1, 2), none of them concludes that "infection of *P. multocida* displays host specificity". Instead, both of them suggest that there is little or no host specificity for *P. multocida* infection.
 15. In figure 2, histological damages of lungs from healthy animals should be included.
- The writing should be carefully checked. For example, line 369 should be "suggest" rather than "suggested",...

Reviewer #2 (Comments for the Author):

Major issue

1. The bacterial strains isolated from the breeding farms were divided into three distinct branches using MLST (Multi-Locus Sequence Typing). To establish whether these strains are responsible for the outbreak and transmission of the disease in rabbits, it is necessary to conduct retrospective experimental studies involving different MLST types to confirm the pathogenicity of the strains leading to fatal outbreaks in rabbits.
2. The article discusses antibiotic resistance genes, but there is no mention of drug sensitivity testing performed on the isolated strains. It should be noted that the genotype cannot completely represent the drug sensitivity phenotype.
3. The article lacks statistical calculations.

Minor issues

1. line 30 serogroup spelling not correct .
2. The majority of the text contains multiple spelling serogroup errors.
3. Line 75 pathogenicity
There are numerous spelling errors throughout the entire text
4. Line 269 Drug susceptibility testing of the isolated strains needs to be conducted."
5. Line 350 The discussion section should be concise and focused, summarizing the key points of this study in a logically coherent manner, while highlighting its main findings.

**Pathogenic and genomic characterisation of rabbit sourced *Pasteurella multocida***
**serogroup F isolates recovered from dead rabbits with respiratory disease**

Jinxiang Wang,^{1,2} Shikun Sun,^{1,2} Dongjin Chen,^{1,2} Chenfang Gao,^{1,2} Lei Sang,^{1,2}

Xiping Xie^{1,2}

**Correspondence:** Wang J (wjx841227@126.com) and Xie X (xpx702@163.com)

**ABSTRACT** *Pasteurella multocida* serogroup F is usually recognized as an
avian-adapted pathogen. However, this serogroup is also widespread in mammals and

[revised manuscript text omitted]

**Virulence genes determination**

Twelve virulence genes (*ptfA*, *tadD*, *pfhA*, *toxA*, *fur*, *tbpA*, *hgbB*, *nanB*, *pmHAS*,
*ompA*, *ompH* and *oma87*) that contribute to the pathogenicity of the *P. multocida* were
screened in the isolates. The virulence genes *ptfA*, *tadD*, *pfhA*, *toxA*, *fur*, *nanB*,
*pmHAS*, *ompA*, *ompH* and *oma87* were screened as described by Tang et al., (19), and
the virulence genes *tbpA* and *hgbB* were screened as described by Ewers et al., (20).
The expected PCR product was purified and sequenced, and the identity of the
product was confirmed by comparing the sequence against the NCBI database.

**Animal experiments**

Rabbits and white feather broilers were used to determine the pathogenicity of the
isolate. The rabbits and white feather broilers were purchased from local farms.
Before infection, nasal, conjunctival and rectal swabs of the rabbits as well as
oropharyngeal and cloacal swabs of the broilers were collected for bacteriological
examination to check the *P. multocida* status of the animals (14). The sera of the

rabbits and broilers were also collected for checking the presence of the *P. multocida*
IgG by using an indirect ELISA assay as described by Jaglic et al., (14).

Thirty-day-old rabbits were divided into groups of eight rabbits (four males and
four females) in each group. Each group was housed in an isolated room, and two
rabbits (one male and one female) from the same group were kept in a cage. Rabbits
from subcutaneous group and intranasal group were subcutaneously and intranasally
inoculated with 6.0×10^4 CFU of the isolate suspended in 100 μ L of sterile normal
saline, respectively. Rabbits from intranasal control group and subcutaneous control
group were intranasally and subcutaneously inoculated with 100 μ L of sterile normal
saline, respectively. The bacterial concentration and infection routes of the rabbit
infection experiment were selected according to the previous reports of Jaglic et al.,
(14) and Wang et al., (16).

Twenty-day-old white feather broilers were divided into groups of eight broilers in
each group. Broilers from the same group were kept in a cage and housed in a
separated room. Broilers from intratracheal group were intratracheally inoculated with
6.0×10^4 CFU of the isolate suspended in 100 μ L of sterile normal saline. Broilers
from intratracheal control group were intratracheally inoculated with 100 μ L of sterile
normal saline. The infection route and bacterial concentration of the broiler infection
experiment were selected as described in the previous reports of Peng et al., (21) and
Jaglic et al., (22), respectively.

The rabbits and broilers had free access to feed and water during the 15-day
experiment period, and the rabbits and broilers were monitored for the clinical signs

of infection during the entire experiment period. All of the rabbits and the broilers
were examined for the presence of pathological lesions at the end of the experiment.
Tracheas, lungs, livers, hearts, spleens, kidneys and whole blood of the all rabbits and
broilers as well as the bursa of Fabricius of the all broilers were collected for
bacteriological examination by using the multiplex PCR assays developed by
Townsend et al., (3) and Harper et al., (18). The lung samples of rabbits from
intranasal group and intranasal control group as well as the lung samples from the all
broilers were used for the histological examination by using hematoxylineosin
staining.

Before inoculation, rabbits (40 mg/kg) and broilers (20 mg/kg) were anaesthetized
by intravenous injection of ketamine to prevent or minimize the sufferings. The
inoculated rabbits and broilers that were caught up in the endpoint of the disease
(dyspnea, loss of appetite or weight loss of 15% or more) were euthanized by bleeding
from the jugular vein under ketamine narcosis. At the end of the experiment, the
survived rabbits and broilers were also sacrificed by bleeding from the jugular vein
under ketamine narcosis.

**Genome sequencing and comparative analyses**

The genomic sequence of the isolates was determined by using the Oxford
Nanopore Technologies (ONT) at Tianjin Novogene Bioinformatic Technology Co.,
Ltd. (Tianjing, China). The raw reads generated by ONT were analyzed by using the
NanoPlot (version 1.30.1), and then the genomic sequence of the isolate was *de novo*
assembled by using the Unicycler (version 0.4.8). The annotation of the genomic

sequence was performed by using the NCBI Prokaryotic Genome Annotation Pipeline.
The prophage, genomic island and drug resistance gene in the genome were
determined by using the web servers of PHAST (<http://phast.wishartlab.com/>),
IslandViewer 4 (<https://www.pathogenomics.sfu.ca/islandviewer/>) and the
comprehensive antibiotic resistance database (<https://card.mcmaster.ca/>), respectively.
To perform the comparative analyses, whole genomic sequences of *P. multocida*
serogroup A strain CQ2 (CP033599), serogroup D strain HN06 (CP003313) and
serogroup F strains including Pm70 (AE004439), HN07 (CP007040), s4 (CP084165),
CIRMBP-0873 (CP020347), CIRMBP-0884 (CP020345), 9N (CP028927), AH09
(CP090521), SD11 (CP090520) and 36502 (CP097792) were freely obtained from
NCBI database. Comparative genome analyses were performed by using the Mauve
(version 2.4.0) (23), BLAST (version 2.4.0) (24), BRIG (version 0.95-dist) (25) and
Easyfig (version 2.2.5) (26).

**RESULTS**

**Capsular and LPS genotyping**

The capsular and LPS genotyping multiplex PCR assays produced the segments of
approximate 850 bp and 470 bp in length for the 19 isolates, respectively. The
sequences of segments from the capsular and LPS genotyping multiplex PCR assays
shared highest identity (up to 99.90% identity) with the *fcbD* and *gatF* of *P. multocida*,
respectively. The results suggested that the serotypes of the 19 isolates were F:L3
(Table S1).

**Multi-locus sequence typing**

The 19 isolates were typed into 4 sequence types by using the multi-host
multi-locus sequence typing (MLST) scheme. ST191 was the most dominant
sequence type (9/19), followed by ST12 (5/19), ST193 (3/19) and ST192 (2/19)
(Table S1). The 19 isolates were typed into 3 sequence types by using the RIRDC
MLST scheme. ST428 was the most dominant sequence type (9/19), followed by
ST430 (8/19) and ST431 (2/19) (Table S1). The 19 isolates were grouped into 3
clusters based on the concatenated housekeeping genes of the multi-host MLST and
RIRDC MLST schemes, and the 19 isolates were not closely related to the *P.*
*multocida* serogroup F strains isolated from chicken (Pm70 and 36502), bovine (F),
swine (HN07) and rodent (9N) (Fig. 1).

**Virulence genes determination**

The virulence genes of *ptfA*, *tadD*, *pfhA*, *fur*, *hgbB*, *ompA*, *ompH* and *oma87* were
positive for the 19 isolates, whereas the virulence genes of *toxA*, *tbpA*, *nanB* and
*pmHAS* were negative for the 19 isolates. By comparing against the NCBI database,
the sequences of the virulence genes (*ptfA*, *tadD*, *pfhA*, *fur*, *hgbB*, *ompA*, *ompH* and
*oma87*) amplified from the 19 isolates shared highest identity (ranged from 99.90 to
100%) with the corresponding genes of *P. multocida*.

**Animal experiments**

Before inoculation, all of the rabbits and broilers were confirmed to be free of *P.*
*multocida* and negative for IgG antibody against *P. multocida*. Rabbits and broilers
from control groups remained free of *P. multocida* and negative for *P. multocida* IgG

at the end of the experiment, and no clinical signs and pathological lesions were
observed in these animals.

All 19 isolates caused severe disease in the subcutaneously inoculated rabbits, and
most of the inoculated rabbits (ranged from 50% (4/8) to 87.5% (7/8)) were
euthanized during the 15-day experiment period (Table S2). Rabbits that were
euthanized within 24 h post inoculation became seriously ill, and diffuse hemorrhagic
pneumonia (Fig. 2A) and diffuse subcutaneous hemorrhage around the inoculation
site (Fig. 2B) were observed in these rabbits. Rabbits that were euthanized after 24 h
post inoculation showed fibrino-hemorrhagic pleuropneumonia (Fig. 2C) and diffuse
subcutaneous abscess around the inoculation site (Fig. 2D). Rabbits that survived the
experiment showed hemorrhagic pneumonia (Fig. 2E) and local subcutaneous abscess
at the inoculation site (Fig. 2F). All of the 19 isolates could be detected in tracheas,
lungs and livers from the subcutaneously inoculated rabbits, and the presence of the
isolates in hearts (12/19), spleens (15/19), kidneys (14/19) and whole blood (12/19)
from the subcutaneously inoculated rabbits could also be detected. Interestingly,
rabbits that survived the subcutaneous inoculation were positive for *P. multocida* IgG
at the end of the experiment.

The intranasal inoculation of the all 19 isolates caused severe disease in the
inoculated rabbits, but no acute disease in these rabbits. Clinical signs including
cough, nasal discharge and dyspnea could be observed in the inoculated rabbits, and a
high proportion of inoculated rabbits (ranged from 37.5% (3/8) to 75% (6/8)) were
euthanized during the 15-day experiment period (Table S2). Rabbits that were

euthanized showed fibrino pleuropneumonia (Fig. 3A) or fibrinopurulent
pleuropneumonia (Fig. 3B), whereas rabbits that survived the experiment showed
pulmonary consolidation with hemorrhagic pneumonia (Fig. 3C). Inflammatory
exudates in bronchiole and alveoli, penetration of red blood cells in bronchiole and
alveoli as well as degeneration of the alveolar epithelial cells were observed in the
lungs showing fibrino pleuropneumonia or fibrinopurulent pleuropneumonia (Fig. 3D).
Inflammatory exudates in bronchiole and alveoli and proliferation of alveolar
epithelial cells were observed in the lungs showing pulmonary consolidation with
hemorrhagic pneumonia (Fig. 3E). All of the 19 isolates could be detected in the
tracheas, lungs and livers from the intranasally inoculated rabbits, and the isolates
could also be detected in hearts (9/19), spleens (13/19), kidneys (8/19) and whole
blood (10/19) from the intranasally inoculated rabbits. Moreover, rabbits that survived
the intranasal inoculation were positive for IgG against *P. multocida*.

All of the intratracheally inoculated broilers survived the challenge showing no
clinical signs of the infection. At the end of the experiment, no pathological lesions
were observed in the inoculated broilers (Fig. S1). The tissue samples including
tracheas, lungs, livers, hearts, spleens, kidneys, blood and bursa of Fabricius from
inoculated broilers were negative for *P. multocida*, the *P. multocida* IgG was also
negative for these broilers, and no histopathological changes were observed in the
lungs of the inoculated broilers (Fig. S1).

**Genome sequencing and comparative analyses**

The genome sizes of the 19 isolates ranged from approximate 2.46 to 2.49 Mbp,

with the average G+C content between 40.3% and 40.35% (Table S3). The predicted
genes of the 19 isolates ranged from 2409 to 2473, with the protein coding sequences
between 2288 and 2356 (Table S3). All of the 19 isolates contained prophages and
genomic islands, and most of the genomic islands were located in the prophage
regions (Table S4). All of the 19 isolates contained drug resistance genes, and the drug
resistance gene profiles of the 19 isolates were quite similar. Twelve isolates including
PF1, PF2, PF3, PF4, PF5, PF6, PF7, PF8, PF10, PF11, PF18 and PF19 contained one
cephalosporin resistance gene and two elfamycin resistance genes, the remaining
seven isolates including PF9, PF12, PF13, PF14, PF15, PF16 and PF17 contained one
cephalosporin resistance gene and one elfamycin resistance gene.

[revised manuscript text omitted]

Discussion

*P. multocida* serogroup F was recognized as the typical avian-adapted pathogen
(5-7). Nevertheless, a relatively high incidence of this serogroup in some mammals

(swine, bovine, sheep and rabbits) has also been documented (8-12). Phylogenetic
analyses using the multi-host MLST housekeeping genes showed that *P. multocida*
serogroup F strains isolated from swine (HN07) (21) and rabbits (16) (s4,
CIRMBP-0873 and CIRMBP-0884) were closest relative to the avian sourced *P.*
*multocida* serogroup F strain Pm70, which suggested that the lack of genetic diversity
among these isolates of different hosts. In contrast to the previous findings, by using
the multi-host MLST and RIRDC MLST housekeeping genes, this study showed that
the rabbit sourced strains (s4, CIRMBP-0873 and CIRMBP-0884) and the pig sourced
strain (HN07) were not closely related to the avian sourced strain Pm70. Moreover, all
of the *P. multocida* serogroup F strains (isolated from swine, bovine, avian, rabbits
and rodent) deposited in the NCBI database were grouped into four distinct clusters.
The 19 rabbit sourced *P. multocida* serogroup F isolates in this study were typed into
4 and 3 sequence types by using multi-host MLST and RIRDC MLST schemes,
respectively. Interestingly, *P. multocida* serogroup F isolates of different sequence
types or pulsed-field gel electrophoresis types were also recovered from rabbits in
Iberian Peninsula (11), Italy (12) and Czech Republic (30). Taken together, these
findings suggested that the genetic diversity of *P. multocida* serogroup F.

Previous studies showed that *P. multocida* serogroup F was pathogenic for rabbits
(14, 16, 22, 31). The rabbit sourced *P. multocida* serogroup F strain J-4103 caused
severe pathogenic lesions and high mortality in intranasally and subcutaneously
challenged rabbits (14). The avian sourced *P. multocida* serogroup F strains P-4218
(turkey) and C21724H3km7 (chicken) could also cause severe diseases and

significant mortality in challenged rabbits (22). Furthermore, the pig sourced *P.*
*multocida* serogroup F strain HN07 was also highly pathogenic for rabbits, which
could cause the death of the challenged rabbits within 48 h post inoculation (31).
However, our previous study showed that the rabbit sourced *P. multocida* serogroup F
strain s4 was a low-virulent strain, which caused mild pathogenic lesions in most of
the challenged rabbits (16). In this study, all of the 19 isolates were highly pathogenic
for rabbits because all of the isolates could result in severe pathogenic lesions and
significant mortality in inoculated rabbits as the previously described high-virulent
strains J-4103 (14), P-4218 (22), C21724H3km7 (22) and HN07 (31) did. Notably,
the infection of J-4103 (14), P-4218 (22) and C21724H3km7 (22) could result in
septicemic disease in challenged rabbits, whereas the 19 isolates in this study and s4
caused purulent disease (16). The results suggested that the various degrees of
pathogenicity of *P. multocida* serogroup F. It should also be noted that rabbit was
highly sensitive to avian and swine sourced *P. multocida* serogroup F strains (22, 31),
and the cross-species transmission of *P. multocida* serogroup F strains from avian and
swine into rabbits might cause significant economic losses in rabbit industry.

*P. multocida* is an important bacterial pathogen that infects a wide range of
animals (1, 2). However, some studies showed that the infection of *P. multocida*
displays host specificity (2, 21). For example, the swine sourced high-virulent *P.*
*multocida* serogroup F strain HN07 that could kill pigs was avirulent to chickens (21).
In this study, all of the 19 rabbit sourced *P. multocida* serogroup F isolates that could
result in severe diseases and high mortality in inoculated rabbits were also avirulent to

white feather broilers. Several studies have tried to explore the molecular basis for
host specificity of *P. multocida* by using comparative genomics (21, 32). Peng et al.,
identified a number of virulence genes in the genomes of avian virulent strains (P1059,
X73 and GX) but absent in that of swine sourced strain HN07 (21), which might
contribute to the avirulence of HN07 to chickens. By comparing the genomes of 13
bovine haemorrhagic septicaemia (HS) strains with those of non-HS strains 36950
(bovine), 3480 (swine), HN06 (swine) and Pm70 (chicken), Moustafa et al., identified
a set of 96 genes unique to the bovine HS strains (32). In this study, several virulence
genes that contribute to fowl cholera were absent in the 19 isolates. Furthermore, by
comparison with the LPS biosynthetic genes of *P. multocida* serotype F:L3 strains
deposited in the NCBI database, polymorphisms were determined in the LPS outer
core biosynthetic genes *natC* and *gatF* among these strains. Interestingly, the LPS
outer core biosynthetic genes *natC* and *gatF* of rabbit sourced strains were identical,
which suggested that the *natC* and *gatF* of the rabbit sourced strains might be selected
by the rabbit immune pressure and associated with the host specificity. In contrast to
these results, a larger-scale comparative genomic analysis based on 109 *P. multocida*
strains of different hosts with multiple types of diseases showed that there were no
genes of *P. multocida* that were specific to a particular type of host (33). Taken
together, the knowledge on the host specificity of *P. multocida* is still limited and
controversial, and the molecular basis for host specificity of the pathogen needs to be
investigated further.

In conclusion, the pathogenicity and genomic features of 19 rabbit sourced *P.*

*multocida* serogroup F isolates were determined in this study. The observations and
findings suggested that the pathogenicity variability and genetic diversity of *P.*
*multocida* serogroup F. Furthermore, the results of this study would also helpful for
the understanding of the host specificity of *P. multocida*.

**ACKNOWLEDGMENTS**

This work was supported by the Fujian Public Welfare Project (2022R10260012),
Construction of Science and Technology Innovation Team of Fujian Academy of
Agricultural Sciences (CXTD2021007-2), 5511 Collaborative Innovation Project of
Fujian Academy of Agricultural Sciences (XTCXGC2021008) and the Earmarked
Fund for China Agriculture Research System (CARS-43-G-5).

**AUTHOR AFFILIATIONS**

¹Institute of Animal Husbandry and Veterinary Medicine, Fujian Academy of
Agricultural Sciences, Fuzhou, Fujian, People's Republic of China

²Fujian Key Laboratory of Animal Genetics and Breeding, Fuzhou, Fujian, People's
Republic of China

**AUTHOR ORCIDs**

Wang J: <http://orcid.org/0000-0001-8911-3068>

**FUNDING**

Funder	Grant(s)	Author(s)
Fujian Public Welfare Project	2022R10260012	Wang J
Construction of Science and Technology Innovation Team of Fujian Academy of Agricultural Sciences	CXTD2021007-2	Wang J
5511 Collaborative Innovation Project of Fujian Academy of Agricultural Sciences	XTCXGC2021008	Wang J
Earmarked Fund for China Agriculture Research System	CARS-43-G-5	Xie X

**AUTHOR CONTRIBUTIONS**

Wang J and Xie X conceived the study. Wang J, Sun S, Chen D, Gao C and Sang
L performed the experiments. Wang J drafted the manuscript. All authors read and
approved the final manuscript.

**DATA AVAILABILITY**

The complete genome sequences of the 19 isolates used in this study were
deposited in the NCBI GenBank (<https://www.ncbi.nlm.nih.gov/genbank/>): PF1
(CP112898, CP112899), PF2 (CP111081), PF3 (CP111082), PF4 (CP111083), PF5
(CP111142), PF6 (CP111143), PF7 (CP111144), PF8 (CP113236), PF9 (CP111145),
PF10 (CP111146), PF11 (CP111147), PF12 (CP112891), PF13 (CP113522,

CP113523), PF14 (CP112892), PF15 (CP112893), PF16 (CP112894), PF17
(CP112895), PF18 (CP112896) and PF19 (CP112897). The datasets used and/or
analysed during the current study are available from the corresponding author on
reasonable request.

**ADDITIONAL FILES**

The following material is available online

Supplemental Material

Supplemental tables: Tables S1, S2, S3 and S4.

Supplemental figures: Figures S1, S2, S3, S4.

**REFERENCES**

[revised manuscript text omitted]

Dear Editors and Reviewers:

Thank you very much for your comments of our manuscript entitled "Pathogenic and genomic characterisation of rabbit sourced *Pasteurella multocida* serogroup F isolates recovered from dead rabbits with respiratory disease" (Manuscript Number: Spectrum03654-23). These comments are all valuable and very helpful for revising and improving our manuscript, as well as the important guidance significance to our research. We have studied the comments carefully and have made corrections. We hope that all the corrections will meet with your approval. The changes were highlighted in the modified manuscript. The main corrections in the paper and the responses to the reviewers' comments are as following:

Reviewer #1 (Comments for the Author):

In this study, the authors reported the virulence and complete genome sequencing of nineteen rabbit originated *Pasteurella multocida* serogroup F. The authors assessed the virulence of these nineteen strains in both rabbit and chicken models and generated the complete genome sequences using Nanopore technology. The authors also attempted to figure out pathogenesis related genes by performing comparative genomics. However, this study also displayed several points of weakness.

1. One of my great concern is the novelty. What is the difference between this study and previously studies assessing the pathogenicity of rabbit-originating serogroup F strains using the same models (references 11, 12, 14, 16 as the authors cited in the manuscript)

Response: Thank you very much for the comment. In the previous studies, the pathogenicity of rabbit sourced *P. multocida* serogroup F strains J-4103 (high-virulent) and s4 (low-virulent) were evaluated in rabbits. The inoculation of high-virulent J-4103 caused high mortality and resulted in severe pathologic lesions in challenged rabbits (acute septicemic syndrome and extensive hemorrhage in subcutis in subcutaneously challenged rabbits, and fibrinopurulent pleuropneumonia and

extensive hemorrhagic pneumonia in intranasally challenged rabbits). The inoculation of low-virulent s4 caused no death and did not result in severe pathologic lesions in the challenged rabbits (local subcutaneous abscess at the inoculation site in subcutaneously challenged rabbits, and hemorrhagic pneumonia, pulmonary consolidation and weak hemorrhagic pneumonia in intranasally challenged rabbits). By comparison with the low-virulent s4, all of the nineteen isolates in the present study were highly pathogenic for rabbits, which resulted in high mortality and caused severe pathologic lesions in the challenged rabbits. However, the severe pathologic lesions (diffuse subcutaneous abscess around the inoculation site in subcutaneously challenged rabbits, and fibrino pleuropneumonia or fibrinopurulent pleuropneumonia in intranasally challenged rabbits) caused by the nineteen isolates of the present study were distinct from those caused by the high-virulent strain J-4103. These results would be helpful for the understanding of the pathogenicity and virulence variability of *P. multocida* serogroup F (**lines 380-393**).

2. Fig. 1, the authors only performed phylogenetic analysis based on the MLST data, why not generate a tree using the whole genome sequences as those data have been generated using Nanopore sequencing? In addition, strains belonging to the other serogroups should be also included in the MLST tree.

3. Line 109, why both multihost and RIRDC typing methods were used? In the current PubMLST online version, only multihost MLST is recommended.

Response: Thank you very much for the two comments (the second and third comments). According to your comments and previous studies (in these studies the multi-host MLST scheme were applied), the phylogenetic tree of the isolates was re-built based on the seven housekeeping genes of the multi-host MLST scheme, and *P. multocida* strains belonging to the other serogroups were also included in the new phylogenetic tree. In the new tree, isolates including PF2, PF3, PF5, PF7, PF8, PF11, PF17 and PF19 were relative to SD11 (type F, rabbit), s4 (type F, rabbit), Pm70 (type F, avian), HN07 (type F, pig), CIRMBP-0884 (type F, rabbit) and 36502 (type F, chicken), isolates including PF1, PF4, PF6, PF9, PF10, PF12, PF14, PF16 and PF18

were relative to CIRMBP-0873 (type F, rabbit), while isolates PF13 and PF15 were grouped in a cluster without any *P. multocida* serogroup F strains (**lines 109-110, lines 116-118, lines 217-227**).

4. Line 119, you have obtained the complete genome sequences, why not analyze more VFGs?

Response: Thank you very much for the comment. According to your comment, a total of 26 virulence genes were screened in the genomes of the nineteen isolates using BLASTn. These virulence genes including adhesion related proteins (*fimA*, *hsf-1*, *hsf-2*, *pfhA*, *pfhB1*, *pfhB2*, *ptfA* and *tadD*), dermonecrotxin (*toxA*), iron binding proteins (*exbB*, *exbD*, *fur*, *hgbA*, *hgbB*, *tbpA* and *tonB*), sialidases (*nanB* and *nanH*), hyaluronidase (*pmHAS*), outer membrane proteins (*ompA*, *ompH1*, *ompH2*, *ompW* and *oma87*) and superoxide dismutase (*sodA* and *sodC*). The result showed that the virulence genes of *fimA*, *hsf-1*, *hsf-2*, *pfhB2*, *ptfA*, *tadD*, *exbB*, *exbD*, *fur*, *hgbA*, *hgbB*, *tonB*, *nanB*, *nanH*, *ompA*, *ompH1*, *ompH2*, *ompW*, *oma87*, *sodA* and *sodC* were positive for the 19 isolates, whereas the virulence genes of *pfhA*, *pfhB1*, *toxA*, *tbpA* and *pmHAS* were negative for the 19 isolates (**lines 120-126, lines 229-233**).

5. Line 171, more information about the sequencing should be added. For example, how many raw data are generated? how did the low-quality data be removed? what is the criterion for defining the low-quality data? how did the phylogenetic trees be generated? etc.

Response: Thank you very much. According to your comment, detailed information about the sequencing of the nineteen isolates had been added in the modified manuscript (**lines 181-189**).

6. Line 256, it would be surprising that no-IgG was detected. Even the strains were avirulent to chickens, antibodies should be induced.

Response: Thank you very much for the comment. We were also surprised by the result of no-IgG was detected in the inoculated chickens. All of the intratracheally

inoculated chickens survived the challenge showing no clinical signs, and at the end of the experiment there were no gross lesions were observed in these inoculated chickens. The results suggested that all of the nineteen *P. multocida* serogroup F isolates in the present study might avirulent to the inoculated chickens. It was supposed that the avirulent *P. multocida* serogroup F isolates were cleaned by innate immune system of the chickens soon as these isolates were intratracheally inoculated (lines 284-285).

7. Line 272, have you tested the corresponding phenotypes?

Response: Thank you very much for the comment. According to the comment and the comment of reviewer #2, antimicrobial susceptibility test of the nineteen isolates was conducted. The results showed that all of the 19 *P. multocida* serogroup F isolates were sensitive to ceftriaxone, cefotaxime, ciprofloxacin, ofloxacin and rifampin, and all the 19 isolates were resistant to ampicillin. Among the 19 isolates, 12 (12/19, 63.16%), 12 (12/19, 63.16%), 8(8/19, 42.11%) and 3 (3/19, 15.79%) were resistant to clarithromycin, chloramphenicol, telithromycin and tetracycline, respectively (lines 128-135, lines 235-240).

8. Animal tests, both high-dose and low-dose challenged groups should be considered as you did not know the virulence of these strains to rabbits before study. Why only chose this dose for inoculation?

Response: Thank you very much for the comment. Before the present study was carried out, a preliminary experiment was used to evaluate the pathogenicity of the isolates in rabbits. In the preliminary experiment, 6 isolates (randomly selected from the nineteen isolates) were selected as the representatives, and rabbits were intranasally infected with the isolate of different doses (1.0×10^{10} , 1.0×10^8 , 1.0×10^6 and 1.0×10^4 CFU). All of the rabbits that infected with the isolate of 1.0×10^{10} and 1.0×10^8 CFU were euthanized within 24 h post-infection because of catching up the endpoint of the experiment (depression, dyspnea, and inability to access feed and water), and all of the rabbits that infected with the isolate of 1.0×10^6 CFU were

ethanized within 48 h post-infection because of catching up the endpoint of the experiment (depression, dyspnea, and inability to access feed and water). The infection of the isolate of 1.0×10^4 CFU did not result in acute disease in the rabbits, and the infected rabbits developed clinical signs and gross lesions, which were identical with those of naturally infected rabbits. Based on the results and the previous studies (in these studies the infection dose of 6.0×10^4 CFU was used), the infection dose of 6.0×10^4 CFU was used in the present study (lines 151-153, lines 159-161).

9. Lines 297-299, while comparative genomic analysis being performed, no useful information is available. Why these nineteen strains were virulent to rabbits? Did they carry any virulence associated genes that were absent from the avirulent strain S4?

Response: Thank you very much for the comment. The previous study showed that the s4 was a low-virulent *P. multocida* serogroup F strain.

The inoculation of s4 did not cause death and result in severe gross lesions in the challenged rabbits (local subcutaneous abscess at the inoculation site in subcutaneously challenged rabbits, and hemorrhagic pneumonia, pulmonary consolidation and weak hemorrhagic pneumonia in intranasally challenged rabbits).

By comparison with the s4, all of the nineteen isolates in the present study were highly pathogenic for rabbits, which resulted in high mortality and severe gross lesions in the challenged rabbits (diffuse subcutaneous abscess around the inoculation site in subcutaneously challenged rabbits, and fibrino pleuropneumonia or fibrinopurulent pleuropneumonia in intranasally challenged rabbits).

To understand why the 19 isolates were highly pathogenic for rabbits, comparative genomic analyses were performed between the 19 isolates and the rabbit sourced low-virulent *P. multocida* serogroup F strain s4. The genomic sizes of the 19 isolates (ranged from 2.46 to 2.49 Mbp) were larger than that of s4 (the genomic size of s4 is approximately 2.06 Mbp), and specific sequences were determined in the genomes of the 19 isolates by comparison with that of s4. About 200 functional genes were determined in the specific sequences, and these functional genes were involved in many vital bacterial physiological processes such as genetic information processing,

environmental information processing, lipid metabolism, amino acid metabolism and energy metabolism, and these functional genes might contribute to the fitness and invasion of the *P. multocida* serogroup F isolates in rabbits (**lines 314-319**).

10. Line 300, remove unexpectedly. It has been well known that *P. multocida* strains recovered from non-avian species generally do not cause fowl cholera.

Response: Thank you very much for the comment. The word “unexpectedly” has been removed (**line 320**).

11. Lines 320-324, several deletions or redundant bases were identified. Did these deletions and redundant bases display any impacts on the biosynthesis or completeness of the LPSs of these nineteen serogroup F strains ?

12. Lines 329-348, did these differences affect the virulence of the nineteen serogroup F strains ?

Response: Thank you very much for the two comments (the 10th and 11th comments). In the present study, in-frame deletion in the LPS outer core biosynthetic gene *natC* and N-terminal redundancy in the LPS outer core biosynthetic gene *gatF* were determined in the 19 rabbit sourced *P. multocida* serogroup F isolates by comparison with those of Pm70 (chicken sourced). It is interesting to elucidate the effects of the deletion and redundancy on the completeness of the LPS, the virulence and host range of the nineteen rabbit sourced serogroup F isolates. For the past several months, we have constructed the plasmids that used to generate the mutants in the genes of *natC* and *gatF*. In the following months, we are going to construct the mutants in the genes of *natC* and *gatF*, and the effects of the deletion and redundancy on the completeness of the LPS, the virulence and host range of the isolates will be evaluated.

13. Line 351, I do not think serogroup F is "typical avian-adapt" as this serogroup has been widely characterized in different host species.

Response: Thank you very much. According to your comment, the phrase "typical avian-adapt" has been removed (**lines 23, 380**).

14. Line 392, I have read these two articles (reference 1, 2), none of them concludes that "infection of *P. multocida* displays host specificity". Instead, both of them suggest that there is little or no host specificity for *P. multocida* infection.

Response: Thank you very much. According to your comment, the two articles (reference 1, 2) had been removed as the references here (**line 394**).

15. In figure 2, histological damages of lungs from healthy animals should be included. The writing should be carefully checked. For example, line 369 should be "suggest" rather than "suggested",...

Response: Thank you very much. According to your comment, histological damages of lungs from healthy animals had been included in figure 2 (**lines 611-612, lines 619-620**), and the writing had also been carefully checked.

Reviewer #2 (Comments for the Author):

Major issue

1. The bacterial strains isolated from the breeding farms were divided into three distinct branches using MLST (Multi-Locus Sequence Typing). To establish whether these strains are responsible for the outbreak and transmission of the disease in rabbits, it is necessary to conduct retrospective experimental studies involving different MLST types to confirm the pathogenicity of the strains leading to fatal outbreaks in rabbits.

Response: Thank you very much for the comment. The previous studies showed that *P. multocida* serogroup F strains isolated from turkey (P-4218), chicken (C21724H3km7) and pig (HN07) were highly pathogenic for rabbits. Based on these results, the present study inferred that the possibility of cross-species transmission of *P. multocida* serogroup F strains from avian and swine into rabbits, which might

cause significant economic losses in rabbit industry. According to your comment, the inference was non-rigorous, and the inference was deleted in the modified manuscript. The comment would be the important guidance significance to our future work, in which the pathogenicity of the strains of different MLST types that responsible for the outbreak and transmission of the disease in rabbits will be evaluated (**lines 380-393**).

2.The article discusses antibiotic resistance genes, but there is no mention of drug sensitivity testing performed on the isolated strains. It should be noted that the genotype cannot completely represent the drug sensitivity phenotype.

Response: Thank you very much for the comment. According to the comment and the comment of reviewer #1, antimicrobial susceptibility test of the nineteen isolates was conducted. The results showed that all of the 19 *P. multocida* serogroup F isolates were sensitive to ceftriaxone, cefotaxime, ciprofloxacin, ofloxacin and rifampin, and all the 19 isolates were resistant to ampicillin. Among the 19 isolates, 12 (12/19, 63.16%), 12 (12/19, 63.16%), 8(8/19, 42.11%) and 3 (3/19, 15.79%) were resistant to clarithromycin, chloramphenicol, telithromycin and tetracycline, respectively (**lines 128-135, lines 235-240**).

3.The article lacks statistical calculations.

Response: Thank you very much for the comment. The statistical calculation has been added in the modified manuscript (**lines 205-206**).

Minor issues

1.line 30 seroproup spelling not correct .

Response: Thank you very much for the comment. We had replaced the “seroproup” with “serogroup”.

2.The majority of the text contains multiple spelling seroproup errors.

Response: Thank you very much for the comment. We had checked the “seroproup”

throughout the entire manuscript and replaced it with “serogroup”.

3. Line 75 pathogenicity

There are numerous spelling errors throughout the entire text.

Response: Thank you very much for the comment. We had checked the “pathogenicity” throughout the entire manuscript and made corrections.

4. Line 269 Drug susceptibility testing of the isolated strains needs to be conducted."

Response: Thank you very much for the comment. According to the comment and the comment of reviewer #1, antimicrobial susceptibility test of the nineteen isolates was conducted. The results showed that all of the 19 *P. multocida* serogroup F isolates were sensitive to ceftriaxone, cefotaxime, ciprofloxacin, ofloxacin and rifampin, and all the 19 isolates were resistant to ampicillin. Among the 19 isolates, 12 (12/19, 63.16%), 12 (12/19, 63.16%), 8(8/19, 42.11%) and 3 (3/19, 15.79%) were resistant to clarithromycin, chloramphenicol, telithromycin and tetracycline, respectively (**lines 128-135, lines 235-240**).

5. Line 350 The discussion section should be concise and focused, summarizing the key points of this study in a logically coherent manner, while highlighting its main findings.

Response: Thank you very much. The discussion section had been re-written according to your comment, and we hope that these corrections will meet with your approval (**lines 371-423**).

Re: Spectrum03654-23R1 (Pathogenic and genomic characterisation of rabbit sourced Pasteurella multocida serogroup F isolates recovered from dead rabbits with respiratory disease)

Dear Dr. Jinxiang Wang:

Thank you for the privilege of reviewing your work. Below you will find my comments, instructions from the Spectrum editorial office, and the reviewer comments.

Reviewer's comment # 3 regarding the lack of statistical analysis has not been fully addressed. Please add more details under your newly added section on the statistical analysis as to what specific tests and what significance thresholds were used.

Revision Guidelines

Sincerely,
Artem Rogovskyy
Editor
Microbiology Spectrum

Dear Editors and Reviewers:

Thank you very much for your comments of our manuscript entitled "Pathogenic and genomic characterisation of rabbit sourced *Pasteurella multocida* serogroup F isolates recovered from dead rabbits with respiratory disease" (Spectrum03654-23R1). We have studied comments carefully and have made corrections that we hope meet with approval. The main corrections in the manuscript and the responds to the reviewer's comments are as following:

Reviewer's comment # 3 regrading the lack of statistical analysis has not been fully addressed. Please add more details under your newly added section on the statistical analysis as to what specific tests and what significance thresholds were used.

Response: Thank you very much. The details of the statistical analysis had been added according to your comment.

All data were analyzed by using the Microsoft Office Excel 2010 (Microsoft Corporation, the USA), and then the statistical summaries were generated. The *t*-test was performed to evaluate the differences between the infected rabbits and the control rabbits, and the *p*-value less than 0.05 was considered to be statistically significant **(lines 227-230)**.

Re: Spectrum03654-23R2 (Pathogenic and genomic characterisation of rabbit sourced Pasteurella multocida serogroup F isolates recovered from dead rabbits with respiratory disease)

Dear Dr. Jinxiang Wang:

Your manuscript has been accepted, and I am forwarding it to the ASM production staff for publication. Your paper will first be checked to make sure all elements meet the technical requirements. ASM staff will contact you if anything needs to be revised before copyediting and production can begin. Otherwise, you will be notified when your proofs are ready to be viewed.

Sincerely,
Artem Rogovskyy
Editor
Microbiology Spectrum